# Glycolytic Reprogramming in Silica-Induced Lung Macrophages and Silicosis Reversed by Ac-SDKP Treatment

**DOI:** 10.3390/ijms221810063

**Published:** 2021-09-17

**Authors:** Na Mao, Honghao Yang, Jie Yin, Yaqian Li, Fuyu Jin, Tian Li, Xinyu Yang, Ying Sun, Heliang Liu, Hong Xu, Fang Yang

**Affiliations:** 1Hebei Key Laboratory for Organ Fibrosis Research, School of Public Health, North China University of Science and Technology, Tangshan 063210, China; namao1991@163.com (N.M.); Sir1254705936@163.com (H.Y.); Jessie1010YJ@163.com (J.Y.); LYQEWBAR@163.com (Y.L.); fuyujinjfy@163.com (F.J.); 13933499300@163.com (H.L.); 2Hebei Key Laboratory for Chronic Diseases, Basic Medical College, North China University of Science and Technology, Tangshan 063210, China; tiantian__1997@163.com (T.L.); zwns69618@163.com (X.Y.); sunying@ncst.edu.cn (Y.S.)

**Keywords:** glycolysis, macrophages, silicosis, N-acetyl-seryl-aspartyl-lysyl-proline, inflammation

## Abstract

Glycolytic reprogramming is an important metabolic feature in the development of pulmonary fibrosis. However, the specific mechanism of glycolysis in silicosis is still not clear. In this study, silicotic models and silica-induced macrophage were used to elucidate the mechanism of glycolysis induced by silica. Expression levels of the key enzymes in glycolysis and macrophage activation indicators were analyzed by Western blot, qRT-PCR, IHC, and IF analyses, and by using a lactate assay kit. We found that silica promotes the expression of the key glycolysis enzymes HK2, PKM2, LDHA, and macrophage activation factors iNOS, TNF-α, Arg-1, IL-10, and MCP1 in silicotic rats and silica-induced NR8383 macrophages. The enhancement of glycolysis and macrophage activation induced by silica was reduced by Ac-SDKP or siRNA-*Ldha* treatment. This study suggests that Ac-SDKP treatment can inhibit glycolytic reprogramming in silica-induced lung macrophages and silicosis.

## 1. Introduction

Silicosis is a serious and fatal occupational lung disease and is mainly caused by inhalation of free crystalline silicon dioxide (SiO_2_) or silica dust [1,2]. Chronic inhalation of silica into the lungs of rats leads to activation of lung macrophages and lung inflammation, which promotes a proteolytic phenotype in macrophages and downregulates the elastin level in lungs, eventually leading to fibrosis [3]. Macrophages involved in the proinflammatory response must rapidly provide energy to fuel the inflammatory response, which is accomplished through glycolysis and high lactate production [4]. Glycolysis is a quick way to produce ATP, and fatty acids serve as precursors for the synthesis of inflammatory mediators in macrophages [5].

Studies have shown that fibrotic lung alveolar macrophages augment glycolysis and increase the expression of multiple key glycolytic mediators in both bleomycin and active transforming growth factor (TGF)-β1-induced fibrotic mouse lungs [6]. Augmentation of glycolysis is an early and sustained event during myofibroblast differentiation in lung fibrosis [7]. Furthermore, the interaction between glycolysis and collagen deposition has been demonstrated in TGF-β1-induced lung fibroblasts and silica-treated mice [8,9]. Although it is acknowledged that glycolysis occurs in silicosis, the exact mechanism driving glycolysis in lung macrophages or lung inflammation remains largely unknown.

Our previous studies showed that the mRNA and protein levels of lactate dehydrogenase A (LDHA) were upregulated in rats that inhaled silica [10,11], suggesting a potential role of LDHA in the glycolysis of silicosis. Increased levels of LDHA and its metabolic product, lactate, were found in patients with idiopathic pulmonary fibrosis (IPF). In addition, a profibrotic feedforward loop was found in IPF patients and myofibroblasts, where LDHA produces lactate, lactate decreases pH in the extracellular space and activates TGF-β1, which can further perpetuate fibrotic signaling [12].

N-acetyl-seryl-aspartyl-lysyl-proline (Ac-SDKP) is a naturally present immunoregulatory active peptide, which is generated from the N-terminal sequence of its precursor thymosin β4 (Tβ4) by meprin α [2,13]. Our group found that Ac-SDKP played an anti-fibrotic role in an experimental silicotic model by inhibiting the epithelial-mesenchymal transformation [14,15,16], myofibroblast transformation [17], and macrophage activation and by alleviating pulmonary inflammation [3]. It has been reported that Ac-SDKP inhibits diabetic nephropathy and has anti-fibrotic effects by regulating the glycolytic pathway [18]. However, it is still unclear whether Ac-SDKP can also play a therapeutic role in pulmonary fibrosis by regulating the glycolytic pathway. Therefore, this study explored the regulatory effect of Ac-SDKP on glycolysis in SiO_2_-induced macrophages through in vivo and in vitro experiments.

## 2. Results

### 2.1. Differential mRNA Expression Profiles of Key Glycolytic Enzymes in Silicotic Rat Lungs

Our previous study used RNA-sequencing techniques to screen differential expression of mRNAs in silicotic rats induced by chronic inhalation of crystalline silica particulates [10]. The results showed that certain key glycolytic enzymes were differentially expressed in silicotic rat lungs, suggesting that changes in metabolic characteristics may be an important link in promoting the progression of silicosis (Figure 1).

### 2.2. High Expression of Glycolysis in Rats Exposed to Silica

To determine whether the occurrence and development of experimental pulmonary fibrosis induced by silica are related to glycolysis reprogramming, we performed immunohistochemistry (IHC) staining of LDHA in the lung tissues of the control group and the model group. The results showed that silicon nodules gradually formed in relation to the exposure time of silica to rats, and LDHA was positively expressed in alveolar macrophages and silicon nodules (Figure 2A). Immunofluorescence (IF) staining showed that LDHA was co-expressed with CD68 (a marker of macrophages) in silicotic rats, indicating that the metabolic activity of LDHA could occur in macrophages (Figure 2B). We also examined the expression of other key enzymes in the glycolytic pathway and found that the protein and mRNA expression levels of HK2, pyruvate kinase M2 (PKM2), and LDHA were also increased in rats with silicosis (Figure 2C). These results showed the existence of the abnormal activation of glycolysis metabolism at different stages of the rat silicosis model, which may be related to the high expression of key enzymes.

### 2.3. Enhanced Glycolysis Was the Crucial Metabolic Characteristic of Silicotic Mice

Subsequently, we further investigated whether glycolytic metabolism was also activated in the mouse silicosis model. We analyzed the expression of LDHA in the mouse pulmonary fibrosis model established by silica and found that the positive expression of LDHA was significantly enhanced in the model group (Figure 3A). As shown in Figure 3B, the protein and mRNA levels of HK2, PKM2, LDHA were significantly upregulated in the lung tissue of fibrotic mice.

### 2.4. Silica Treatment Increased the Level of Glycolysis in Macrophages

To investigate the metabolic changes in macrophages, we used the silica-induced macrophage activation model. We found that with the increase in silica concentration, the fluorescence intensity of LDHA gradually increased (Figure 4A). As shown in Figure 4B, silica increased the expression of the key glycolysis enzymes HK2, PKM2, LDHA, and macrophage activation indicators inducible nitric oxide synthase (iNOS), tumor necrosis factor-α (TNF-α), arginase-1 (Arg-1), interleukin-10 (IL-10), and monocyte chemoattractant protein-1(MCP1) in NR8383 cells in a dose-dependent manner. With the increase in silica dose, the concentrations of extracellular lactate increased gradually (Figure 4C). These results suggest that the increased glycolysis induced by silica may be a vital metabolic feature of macrophage activation.

### 2.5. Ac-SDKP Attenuated the Enhancement of Glycolysis in Macrophages Treated with Silica

We previously demonstrated that Ac-SDKP alleviates the inflammatory response of macrophages in silicotic rats [3]. In this study, we explored whether Ac-SDKP exerts anti-inflammatory effects by regulating the energy metabolism of macrophages. We used Ac-ADKP, a functionally inactive analog of Ac-SDKP (replacement of Ser by Ala in Ac-SDKP), as a control to evaluate the effects of Ac-SDKP [3].

IF staining results showed that Ac-SDKP treatment could reduce silica-induced co-expression of MCP1/LDHA and Arg-1/LDHA, whereas Ac-ADKP had no effect (Figure 5A,B). Western blotting (Figure 5C) indicated that silica activated HK2, PKM2, and LDHA expression and increased the expression of iNOS, TNF-α, Arg-1, IL-10, and MCP1 in NR8383 macrophages. Ac-SDKP treatment reversed all SiO_2_-induced effects, whereas Ac-ADKP had no effect. Then, the concentration of lactate in the cell culture medium was measured. The results showed that Ac-SDKP reduced the increase in lactate production induced by silica (Figure 5D). Reactive oxygen species (ROS) are essential for the induction and maintenance of M1/M2 macrophage polarization [19]. We used a fluorescent probe, 2,7-dichlorodihydrofluorescein diacetate (DCFH-DA), to detect whether Ac-SDKP regulated the ROS production in macrophages. The results show that silica significantly increased the intracellular ROS production and was reversed by Ac-SDKP treatment (Appendix A).

### 2.6. Knockdown of Ldha Inhibited Silica-Induced Macrophage Activation

LDHA is a key enzyme in glycolysis regulation. To further investigate the effect of LDHA, we knocked down the expression of *Ldha* using small interfering RNA (siRNA). The results show that siRNA-*Ldha*#3 clearly exhibits silencing effect, and it was used in the subsequent experimental study. After the silica treatment, MCP1 and LDHA or Arg-1 and LDHA were co-expressed in macrophages, and the effects of the silica treatment were inhibited by knockdown of *Ldha* (Figure 6A–C). SiRNA-*Ldha* transfection significantly inhibited the expression of iNOS, TNF-α, Arg-1, IL-10, and MCP1 at baseline and the silica stimulation level (Figure 6D). The analysis of lactate concentration showed similar results (Figure 6E). These results indicate that LDHA is involved in the activation of NR8383 alveolar macrophages. It is documented that LDHA could promote ROS to induce inflammation [20]. Therefore, we examined the effect of LDHA on ROS production. The results show that silica significantly increased the intracellular ROS production and was reversed by siRNA-*Ldha* (Appendix A).

### 2.7. Ac-SDKP Attenuated High Glycolytic Activity and Macrophage Activation in Rats Exposed to Silica 

We further explored the regulatory effect of Ac-SDKP on glycolysis in rats exposed to silica. As shown in Figure 7, key glycolysis enzymes, macrophage activation indicators and lactate concentrations increased in the lung tissue of rats that inhaled silica for 24 weeks. Treatment with Ac-SDKP attenuated the increase in glycolytic signaling, macrophage activation indicators and lactate concentrations in silicotic rats.

## 3. Discussion

Glycolytic reprogramming promotes pulmonary fibrosis because glycolysis is a major energy driving force for myofibroblast differentiation [7] and activation of macrophages [6], both of which contribute to lung inflammation and collagen deposition. Some studies have suggested the potential role of glycolysis in silicosis. However, the role of glycolysis in silicosis is still unclear. The lactate levels and the protein expression levels of glycolytic enzymes, including HK2, phosphofructokinase muscle (PFKM), PKM2, and pyruvate dehydrogenase kinase isozyme1 (PDK1), were enhanced in TGF-β1-induced MRC-5 fibroblasts and in mice exposed to silica [8]. The high levels of HK2 and PKM2 in silicotic mice were attenuated by triiodothyronine (T3) treatment [9]. These studies showed an abnormal level of glycolytic reprogramming in mice exposed to silica or TGF-β1 treatment. In our previous study, we also found that the major glycolytic enzymes were altered in silicotic rats [10]. In the present study, we found increased levels of HK2, PKM2, and LDHA in rats that inhale silica and in silicotic mice. In addition, we found that the expression of LDHA was mostly in lung macrophages, accompanied by enhancement of inflammatory indicators, suggesting that silica promotes inflammation and glycolysis in alveolar macrophages. 

It is well documented that classical activated (M1) macrophages mainly rely on glycolysis to provide energy and play a proinflammatory role; alternatively activated (M2) macrophages preferentially utilize oxidative phosphorylation to promote the repair of tissue damage [21]. Furthermore, bone marrow-derived macrophages mainly rely on glycolysis to participate in the inflammatory response, while tissue-resident macrophages mainly rely on oxidative phosphorylation and are not sensitive to glycolysis [22,23]. However, accumulating evidence suggests that macrophage metabolism is not as simple as previously thought [24]. In this study, we found that the markers of M1 and M2 macrophages all increased in rats exposed to silica for 24 weeks, suggesting lung inflammation regulated by M1 macrophages and a profibrotic effect regulated by M2 macrophages have vital roles in this stage of silicosis. The in vitro study also showed increasing levels of iNOS, TNF-α, Arg-1, IL-10, and MCP1 in NR8383 alveolar macrophages. These results suggest that glycolysis contributes to the regulation of macrophage in silicosis to promote activation of M1/M2 macrophages. 

In previous studies, the mRNA and protein expression levels of LDHA were increased in rats that inhaled silica [10,11]. As the key enzyme in glycolysis, LDHA preferentially converts pyruvate to lactate. The levels of LDHA and its metabolic product, lactate, were increased in patients with IPF, bleomycin-induced mice, and radiation-induced mice, TGF-β-induced lung fibroblasts, and cystic fibrosis lung epithelial IB3-1 cells [25,26,27,28]. FX11, a specific LDHA inhibitor, reduced the lipopolysaccharide (LPS)-activated expression of interleukin-6 (IL-6), iNOS, and cyclooxygenase-2 (COX-2) and suppressed the production of IL-6 and nitrites [27]. In this study, we also found high levels of LDHA and lactate in silicotic models, suggesting the potential effect of LDHA on silicosis. Then, we found that the silencing of *Ldha* inhibited the high levels of iNOS, TNF-α, Arg-1, IL-10, and MCP1 in NR8383 alveolar macrophages induced by silica exposure. These results show that the inhibition of glycolysis reduces silica-induced inflammation. 

Ac-SDKP is an anti-inflammatory tetrapeptide released from thymosin-β4 by the sequential action of meprin-α and prolyl oligopeptidase (POP), and it is cleaved by angiotensin-converting enzyme (ACE) [29]. In preclinical studies, Ac-SDKP protected from a wide range of lung, kidney, and cardiovascular injury models and demonstrated anti-inflammatory properties, decreasing the infiltration by macrophages and neutrophils, nuclear factor kappa-B (NF-κB) activation or inflammatory molecules, as well as antifibrotic effects [29]. Studies have shown that the anti-fibrosis properties of Ac-SDKP are partly mediated by its anti-inflammatory activity, i.e., inhibiting the differentiation of bone marrow stem cells to macrophages, the activation and migration of macrophages, inflammatory signaling pathways, and the release of cytokines [30,31,32]. A recent study also showed that Ac-SDKP disrupts the defective metabolism-linked mesenchymal transformation to exert antifibrotic effect on diabetic kidney, and glycolysis inhibitors could elevate Ac-SDKP to reverse kidney fibrosis [18]. This study showed a potential relationship between Ac-SDKP and glycolysis and documented that the inhibition of glycolysis reduces organ fibrosis by increasing the level of Ac-SDKP. In this study, we found that Ac-SDKP treatment inhibited the enhancement of glycolysis and activation of macrophages induced by silica, suggesting that Ac-SDKP produces anti-inflammatory and anti-fibrotic effects by the inhibition of glycolysis.

## 4. Materials and Methods

### 4.1. Animal Experiments

All procedures used in this study were reviewed and approved by the Committee on the Ethics of North China University of Science and Technology (LX2019033) and complied with the US National Institutes of Health Guide for the Care and Use of Laboratory Animals [3]. Specific pathogen-free (SPF) adult male Wistar rats (160 ± 10 g) were obtained from Beijing Vital River Laboratory Animal Technology Co. Ltd. Rats were housed in SPF conditions with a daily light/dark cycle of 12 h light and 12 h darkness. Water and a standard laboratory diet were given ad libitum [33].

The silicosis rat model was established using the HOPE-MED8050 dynamic dust pollution control system (HOPE Industry and Trade, Tianjin, China) [3,34]. The rats were randomly divided into 3 groups with 10 rats in each group: (1) control group: inhaled pure air for 16 weeks and treated with 0.9% normal saline for 8 weeks; (2) silicosis group: inhaled 50 ± 10 g/m^3^ of silica (s5631; Sigma-Aldrich, St. Louis, MO, USA; ground and then heated at 180 °C for 6 h) daily for 16 weeks, and then treated with 0.9% normal saline for 8 weeks; and (3) Ac-SDKP treatment group: inhaled silica for 16 weeks, and then treated with 800 μg/kg/d Ac-SDKP (H-1156; Bachem AG, Torrance, CA, USA) until 24 weeks. Ac-SDKP or 0.9% saline was delivered at an average flow rate of 0.11 μL/h via mini osmotic pumps (Alzet 2ML, DURECT, Co. Ltd., USA) implanted into the abdominal cavity of rats. Lung tissue samples were collected at the end of the experimental periods. 

Eight-week-old SPF male C57BL/6 mice were purchased from Vital River Laboratory Animal Technology. Mice were randomly divided into two groups (*n* = 6). The control group mice received 50 μL saline instillation, and mice in the silicosis group received a one-time dose of SiO_2_ (5 mg/mouse) [2,35]. 

### 4.2. Cell Culture and Treatments

NR8383 alveolar macrophages was purchased from the Cell Bank of the Chinese Academy of Sciences (Shanghai, China) and cultured in 25 cm^2^ flasks in Ham’s F-12K medium (L450KJ; Shanghai BasalMedia Technologies Co., Ltd. Shanghai, China) [2]. After serum starvation for 24 h, 10 nmol/L Ac-SDKP or 10 nmol/L Ac-ADKP (PCM10745-0912; Pepmic, Suzhou, China) [3] was added 1 h prior to SiO_2_ treatment and then the cells were co-treated for another 24 h in serum-free medium.

### 4.3. Cell Transfection

Transient transfections were carried out using Lipofectamine 2000 reagent (2319729, Invitrogen, USA) according to the manufacturer’s instructions [1,36]. The silencing sequences of *Ldha* were designed and synthesized by RiboBio Co., Ltd. (Guangzhou, China). Cells were subjected to serum starvation for 1 h and then transfected with siRNAs targeting *Ldha* or the negative siRNA control. The target sequences of the siRNA-*Ldha* were derived from following sequences: (1) siRNA-*Ldha* GCTTGTGCCATCAGTATCT; (2) siRNA-*Ldha* GGGAGAGATGATGGATCTT; and (3) siRNA-*Ldha* TCCCATTTCCACCATGATT.

### 4.4. Immunohistochemistry and Immunofluorescence Staining

Immunohistochemistry staining was performed using previously reported protocols [37,38]. The antigen was retrieved using the heat-induced antigen retrieval method, and then 3% hydrogen peroxide was used to quench the endogenous peroxidases. The sections were subsequently incubated with the primary antibody against LDHA (1:100 dilution, DF6280, Affinity) overnight at 4 °C. The following day, the sections were incubated with the secondary antibody (PV-6000, Beijing Zhongshan Jinqiao Bio-technology Co. Ltd, China) at 37 °C for 30 min. Immunoreactivity was visualized using DAB (ZLI-9018; Beijing Zhongshan Jinqiao Biotechnology Co. Ltd.). Cells with brown staining were considered LDHA-positive. Results were visualized by light microscopy. For immunofluorescence staining, paraffin sections of the lung tissue and cell samples were incubated with LDHA, CD68 (ab201340, Abcam)/LDHA, MCP1 (RM01549, ABclonal)/LDHA and Arg-1 (610708, BD)/LDHA overnight at 4 °C. The following day, samples were incubated with the secondary antibody at 37 °C for 40 min. Nuclei were stained with DAPI (8961s; Cell Signaling Technology, Inc., Danvers, MA, USA).

### 4.5. Western Blotting

Western blotting was performed using published protocols [2,3,33] with antibodies directed against HK2 (A01389, Boster), PKM2 (BM4601, Boster), LDHA (DF6280, Affinity), iNOS (ARG56509, arigo), TNF-α (GTX110520, GeneTex), Arg-1 (610708, BD), IL-10 (DF6894, Affinity), MCP1 (A7277, ABclonal), α-tubulin (Tub α, GTX112141, GeneTex), and β-actin (AC026, ABclonal).

### 4.6. Quantitative Real-Time PCR (qRT-PCR)

Quantitative RT-PCR was performed as described in previous studies [1,34]. The primer sequences were as follows: *Hk2* forward 5′-CGCAGAGGGGACTTTGACATT-3′ and reverse 5′-GTTCAGTGAGCCCATGTCGAT-3′; *Pkm* forward 5′-CAGCAACGCTTGTAGTGCTC-3′ and reverse 5′-GAAGCAAAGCCCAGAGATGC-3′; *Ldha* forward 5′-TAACCCAGAACTGGGCACTG-3′ and reverse 5′-ATGGCCCAGGATGTGTAACC-3′; *Gapdh* forward 5′-GGTGAAGGTCGGTGTGAACG-3′ and reverse 5′-CTCGCTCCTGGAAGATGGTG-3′; and *Actb* forward 5′-GTCGTACCACAGGCATTGTGATGG-3′ and reverse 5′-GCAATGCCTGGGTACATGGTGG-3′. The results were calculated using the 2^−^^ΔΔCT^ method.

### 4.7. Lactate Production Measurement

The lactate concentration in the cell culture medium and harvested lung tissue was measured using a colorimetric assay. A lactate assay kit (Nanjing Jiancheng Bioengineering Institute) was used according to the manufacturer’s protocols.

### 4.8. Detection of ROS

The production of ROS was measured by staining cells with a fluorescent probe DCFH-DA (Cayman Chemical, USA) according to the manufacturer’s instructions. The cells were loaded with 10 µmol/L DCFH-DA at 37 °C for 30 min. Then, the cells were washed twice with PBS, and the intensity of DCFH-DA fluorescence was measured using a fluorescence microscope, with an excitation wavelength of 502 nm and an emission wavelength of 523 nm. The mean fluorescence intensity was analyzed using Image-Pro Plus 6.0 (Media Cybernetics, Rockville, MD, USA).

### 4.9. Statistical Analysis

SPSS 20.0 statistical software (SPSS Inc., Chicago, IL, USA) was used to analyze the data. All data were expressed as the mean ± standard deviation (SD). Two-group comparisons were made using unpaired Student’s *t*-test. Multiple group comparisons were performed by one-way analysis of variance followed by Tukey’s post hoc analysis. Results were considered statistically significant when *p* < 0.05 at a 95% confidence interval.

## 5. Conclusions

In summary, we demonstrated that the abnormal activation of glycolysis is accompanied by the process of silicosis, and Ac-SDKP alleviates pulmonary fibrosis by inhibiting the expression of glycolytic signals in macrophages. Blocking *Ldha* antagonizes the occurrence and development of silicosis by inhibiting macrophage activation.

## Figures and Tables

**Figure 1 ijms-22-10063-f001:**
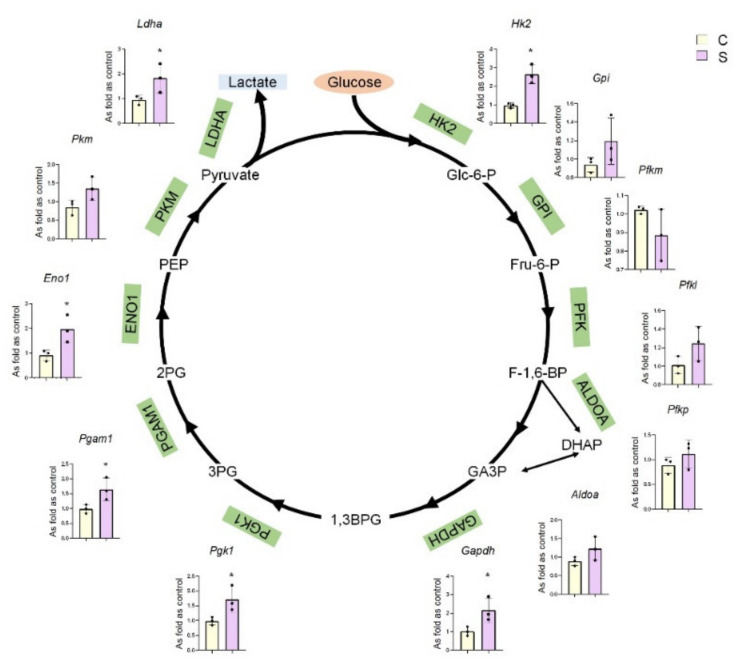
Differential mRNA expression profiles of key glycolytic enzymes in silicotic rat lungs. *Hk2*/HK2: hexokinase2, *Gpi*/GPI: glucose-6-phosphate isomerase, *Pfkm*: phosphofructokinase muscle, *Pfkl*: phosphofructokinase liver type, *Pfkp*: phosphofructokinase platelet, *Aldoa*/ALDOA: fructose-bisphosphate aldolase A, *Gapdh*/GAPDH: glyceraldehyde-3-phosphate dehydrogenase, *Pgk1*/PGK1: phosphoglycerate kinase1, *Pgam1*/PGAM1: phosphoglycerate mutase1, *Eno1*/ENO1: enolase1, *Pkm*/PKM: pyruvate kinase M, *Ldha*/LDHA: lactate dehydrogenase A, PFK: phosphofructokinase. * Compared with control group, *p* < 0.05. C: control 24 w; S: silicosis 24 w.

**Figure 2 ijms-22-10063-f002:**
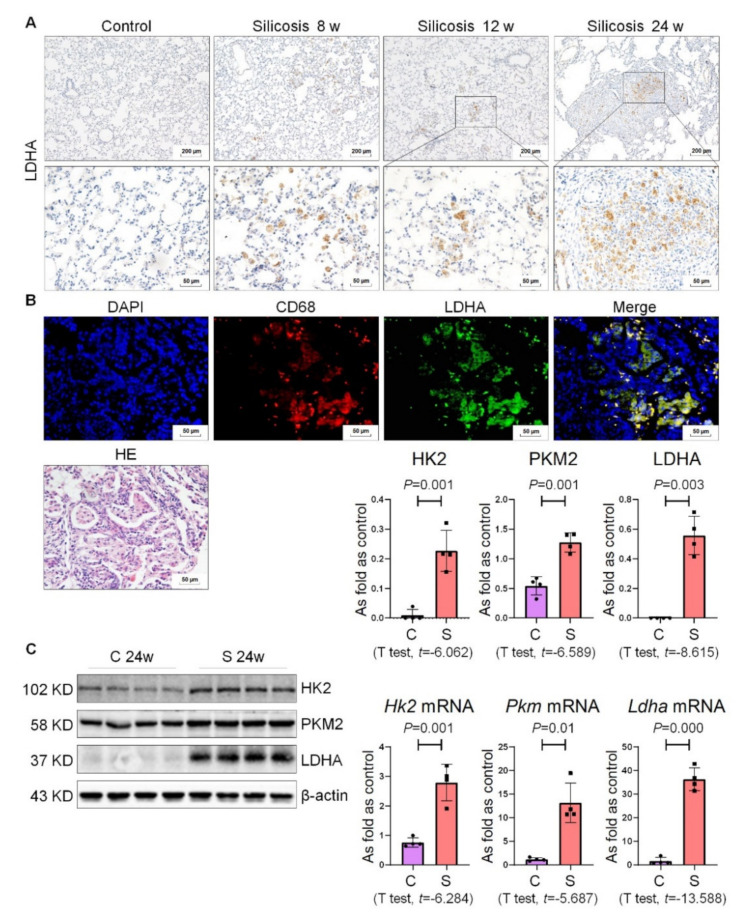
High expression of glycolysis in rats exposed to silica. (**A**) Positive expression of LDHA in silicotic rats observed by IHC staining, bars = 200 μm or 50 μm. (**B**) The co-expression of CD68 and LDHA in silicotic rats was measured by IF staining, bar = 50 μm. (**C**) Protein and mRNA expression levels of HK2, PKM2, and LDHA in silicotic rats measured by Western blotting and qRT-PCR. Data are presented as the mean ± SD. *n* = 4 per group. C: control 24 w; S: silicosis 24 w.

**Figure 3 ijms-22-10063-f003:**
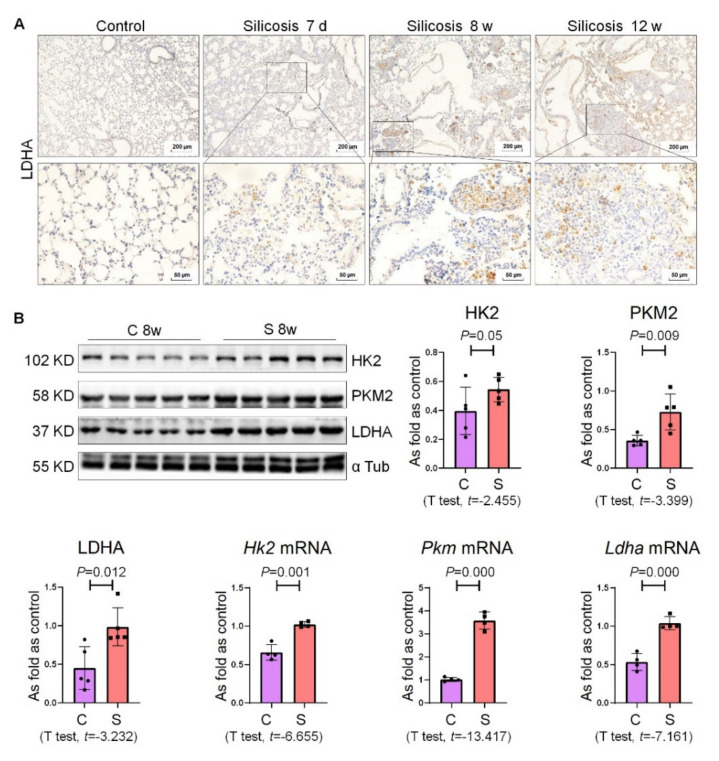
Enhanced glycolysis is the crucial metabolic characteristic of silicotic mice. (**A**) Expression of LDHA in silicotic mice measured by IHC staining, bars = 200 μm or 50 μm. (**B**) Protein and mRNA expression levels of HK2, PKM2, and LDHA in mouse lungs measured by Western blotting and qRT-PCR. Data are presented as the mean ± SD. *n* = 5 or 4 per group. C: control 8 w; S: silicosis 8 w.

**Figure 4 ijms-22-10063-f004:**
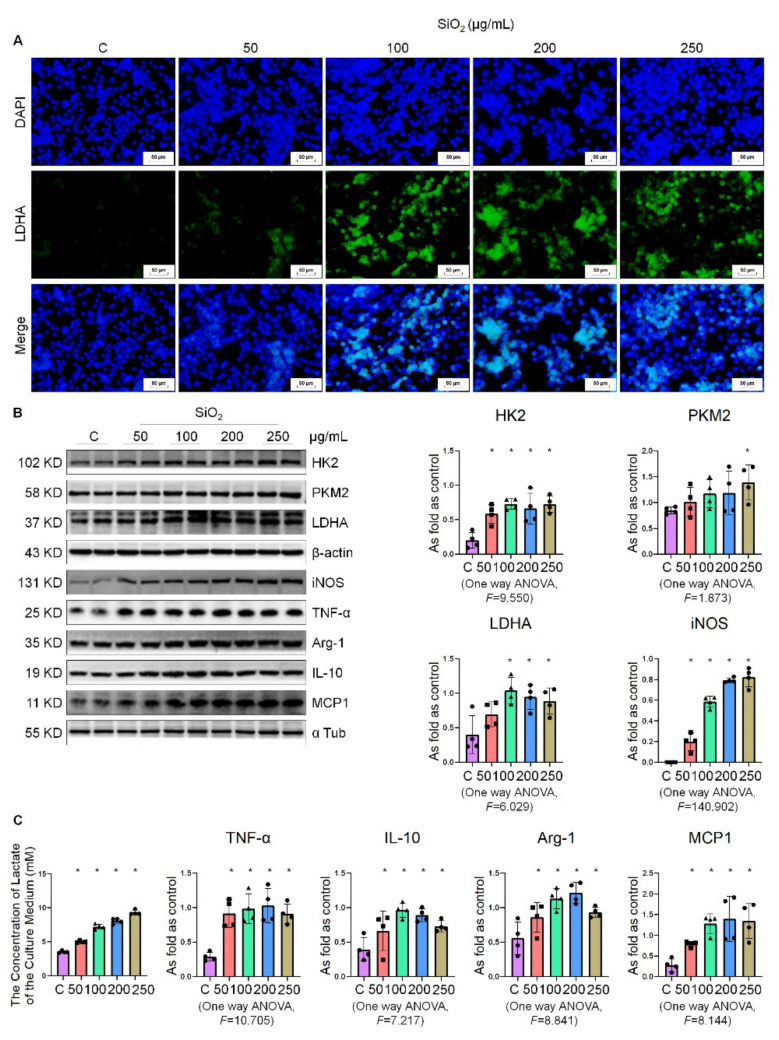
Silica treatment increases the level of glycolysis in macrophages. (**A**) Changes in LDHA expression in NR8383 cells treated with 50, 100, 200, and 250 µg/mL silica measured by IF staining, bar = 50 μm. (**B**) Levels of HK2, PKM2, LDHA, iNOS, TNF-α, Arg-1, IL-10, and MCP1 in NR8383 cells treated with 50, 100, 200, and 250 µg/mL silica measured by Western blotting. * Compared with control group, *p* < 0.05. Data are presented as the mean ± SD. *n* = 4 per group. (**C**) The lactate content in the culture medium was detected using a lactate assay kit. *Compared with control group, *p* < 0.05.

**Figure 5 ijms-22-10063-f005:**
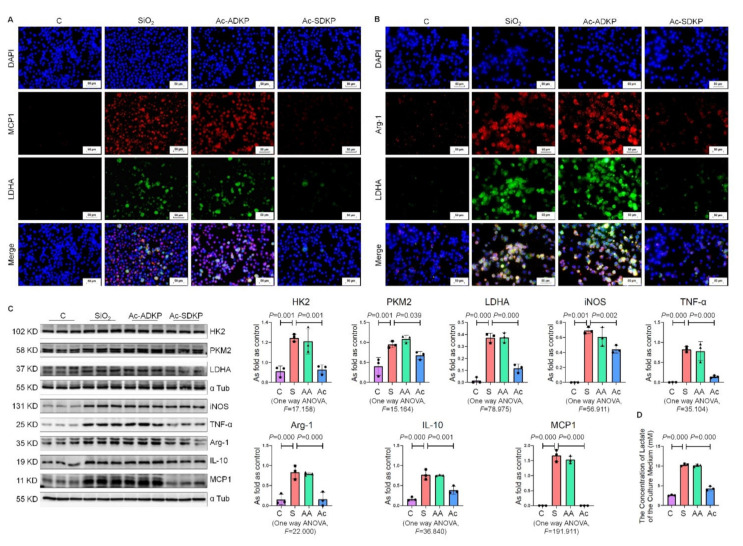
Ac-SDKP attenuates the enhancement of glycolysis in macrophages treated with silica. (**A**) The co-expression of MCP1 and LDHA in NR8383 cells was measured by IF staining, bar= 50 μm; (**B**) The co-expression of Arg-1 and LDHA in NR8383 cells was measured by IF staining, bar= 50 μm. (**C**) Protein expression of HK2, PKM2, LDHA, iNOS, TNF-α, Arg-1, IL-10, and MCP1 in NR8383 cells treated with or without SiO_2_, Ac-ADKP, and Ac-SDKP. Data are presented as the mean ± SD. *n* = 3 per group. (**D**) The lactate content in the culture medium was detected using a lactate assay kit. C: control; S: SiO_2_; AA: Ac-ADKP: SiO_2_ and Ac-ADKP; Ac: Ac-SDKP: SiO_2_ and Ac-SDKP.

**Figure 6 ijms-22-10063-f006:**
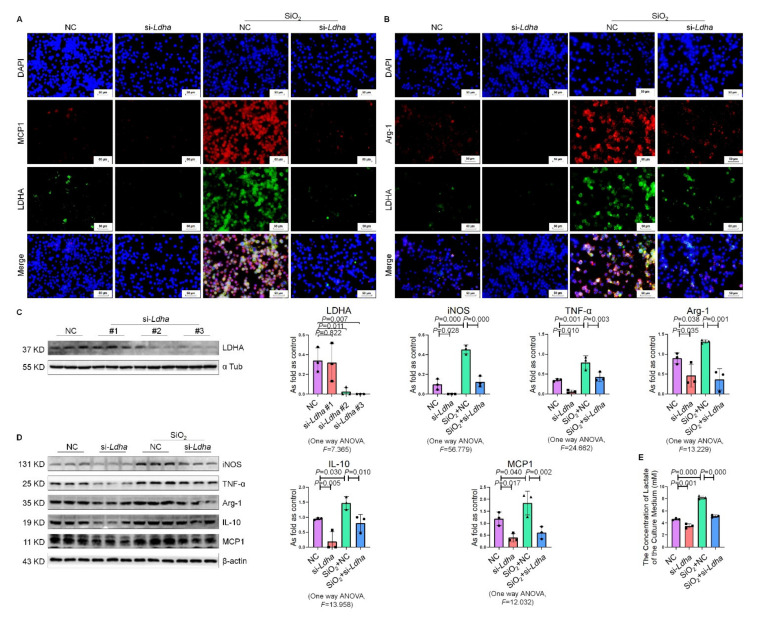
Knockdown of *Ldha* inhibits silica-induced macrophage activation. (**A**) The co-expression of MCP1 and LDHA in NR8383 cells was measured by IF staining, bar = 50 μm. (**B**) The co-expression of Arg-1 and LDHA in NR8383 cells was measured by IF staining, bar =50 μm. (**C**) Transient transfection of siRNA-*Ldha* sequence in macrophages. (**D**) The protein levels of iNOS, TNF-α, Arg-1, IL-10, and MCP1 in NR8383 cells transfected with siRNA-*Ldha* and treated with or without SiO_2_ were measured by Western blotting. Data are presented as the mean ± SD. *n* = 3 per group. (**E**) The lactate content in the culture medium was detected using a lactate assay kit. NC: negative control; si-*Ldha*: siRNA-*Ldha*.

**Figure 7 ijms-22-10063-f007:**
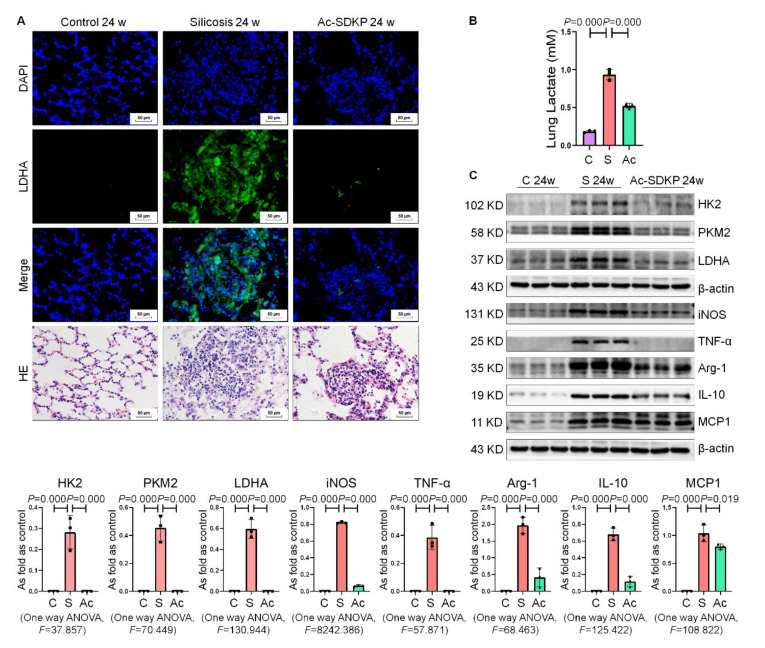
Ac-SDKP attenuates high glycolytic activity and macrophage activation in rats exposed to silica. (**A**) The expression of LDHA in rats exposed to silica was measured by IF staining, bar = 50 μm. (**B**) The lactate content in rat lung tissue was detected using a lactate assay kit. (**C**) Levels of HK2, PKM2, LDHA, iNOS, TNF-α, Arg-1, IL-10, and MCP1 in silicotic rats were measured by Western blotting. Data are presented as the mean ± SD. *n* = 3 per group. C: control 24 w; S: silicosis 24 w; Ac: Ac-SDKP treatment 24 w.

## Data Availability

The underlying data of the study can be obtained by contacting the authors if it is reasonable.

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
