# Peer review of "Glycolytic Reprogramming in Silica-Induced Lung Macrophages and Silicosis Reversed by Ac-SDKP Treatment"

_ijms, 2021, doi:10.3390/ijms221810063_

Round 1
Reviewer 1 Report
In this manuscript, the authors examined the expression of glycolytic enzymes in the lung in silicosis animals. Expression of glycolytic enzymes was elevated in the lung of silicotic mice and silicotic rats. Silica treatment also increased expression of glycolytic enzyme genes in NR8383 alveolar macrophages. Treatment with Ac-SDKP and si-Ldha inhibited silica-induced macrophage activations. Ac-SDKP treatment also attenuated glycolytic enzyme induction and macrophage activation in the lung of silicotic rats. Although the results are clearly presented and are very interesting, I have the following comments and questions.
- Please explain in more detail how Sc-SDKP exhibits anti-inflammatory effect. Is there a membrane receptor for Sc-SDKP? Does Sc-SDKP act in cytoplasm after entering there?
- Are ROS levels increased in the lung of silicotic animals or/and in macrophages treated with silica? If so, can treatment with Ac-SDKP and/or si-Ldha decrease ROS levels?
- Abstract. line 18. "NR8383 cell line was used in vivo experiments". "in vivo" may be "in vitro"?
- Results. Line 134. Please explain Ac-ADKP in detail. What's Ac-ADKP? How is it different from Ac-SDKP?
- Figure 5. Please add explanation about "C", "S", "AA" and "AS" in the legend.
- Figure 7. Please add explanation about "C", "S" and "AS" in the legend.
- Discussion. Please discuss how Ac-ASKP suppresses inflammation and glycolysis in more detail (related to comment #1) and how glycolysis inhibition (such as by si-Ldha) affects macrophage activation and polarization.
- Methods. Please explain the administration route of Ac-SDKP in animal experiments. Inhalation or i.v.?
Author Response
Dear reviewer
Thank you very much for your comments concerning our manuscript, originally titled “Glycolytic reprogramming in silica-induced lung macrophages and silicosis reversed by Ac-SDKP treatment”. All comments were valuable and very helpful for revising and improving our paper, and they have provided important guiding significance for our research. We have carefully considered all of the comments and revised our manuscript accordingly. We hope that the changes we have made will meet with your approval. The revisions to the manuscript were marked up using the “Track Changes” function.
Point 1: Please explain in more detail how Sc-SDKP exhibits anti-inflammatory effect. Is there a membrane receptor for Sc-SDKP? Does Sc-SDKP act in cytoplasm after entering there? 

Response 1: Studies have shown that the anti-fibrosis properties of Ac-SDKP are partly mediated by its anti-inflammatory activity, that is, inhibiting the differentiation of bone marrow stem cells to macrophages, the activation and migration of macrophages, inflammatory signaling pathways, and the release of cytokines.
Despite the localization and characterization of Ac-SDKP binding sites in myocardium (Am J Physiol Heart Circ Physiol 2007, 292 (2), H984-993.), the Ac-SDKP receptors were still not identify or clone. We will try to make antibody for screening the putative receptors of Ac-SDKP in the future.
Point 2: Are ROS levels increased in the lung of silicotic animals or/and in macrophages treated with silica? If so, can treatment with Ac-SDKP and/or si-Ldha decrease ROS levels?
Response 2: We used fluorescent probe 2,7-Dichlorodihydrofluorescein diacetate (DCFH-DA) to measure intracellular ROS production. The results showed that the expression of ROS was significantly enhanced in silica-induced macrophages and reversed by Ac-SDKP or Ldha siRNA treatment. These results was supplemented as Figure S1.
Point 3: Abstract. line 18. "NR8383 cell line was used in vivo experiments". "in vivo" may be "in vitro"?
Response 3: This error has been corrected in the revised manuscript.
Point 4: Results. Line 134. Please explain Ac-ADKP in detail. What's Ac-ADKP? How is it different from Ac-SDKP?
Response 4: We used Ac-ADKP, a functionally inactive analogue of Ac-SDKP (replacement of Ser by Ala in Ac-SDKP), as a control to evaluate the effects of Ac-SDKP. The related context in result and method sections has been revised.
Point 5: Figure 5. Please add explanation about "C", "S", "AA" and "AS" in the legend.
Response 5: We have revised the figure legends.
Point 6: Figure 7. Please add explanation about "C", "S" and "AS" in the legend.
Response 6: We have revised the figure legends.
Point 7: Discussion. Please discuss how Ac-ASKP suppresses inflammation and glycolysis in more detail (related to comment #1) and how glycolysis inhibition (such as by si-Ldha) affects macrophage activation and polarization.
Response 7: We have revised the Discussion section to explain the role of Ac-SDKP on inflammation and glycolysis, and the effect of si-Ldha on macrophage activation and polarization
Point 8: Methods. Please explain the administration route of Ac-SDKP in animal experiments. Inhalation or i.v.?
Response 8: Thank you for carefully reviewing my article. We have supplemented the administration route of Ac-SDKP about animal experiments in Materials and Methods section.
Reviewer 2 Report
In the manuscript, the authors demonstrate that, during silicosis, the metabolic switch toward glycolysis of lung tissue and of alveolar macrophages is responsible for the polarization of macrophages toward the M2 phenotype, and that treatment with the Ac-SDKP compound shows interesting activity on both processes: glycolysis and macrophage polarization. The data are convincing, well organized, and very interesting.
Comments
-the abstract is not clear, acronyms are not necessary, the cell experiments are in vitro and not in vivo, and the results could be enriched by a few more results, e.g. be less general.
-The authors introduce the mouse silicosis model, but on the same they do not evaluate the effect of Ac-SDKP on glycolysis enzymes, nor the switch to an M2 phenotype of macrophages. What is the value added of this model?
-Line 235-236: Thus, the statement” We found that Ac-SDKP treatment attenuates glycolysis in silicotic rats and mice or in silica-induced NR8383 236 alveolar macrophages.” is incorrect, Ac-SDKP was not used to treat the silica-induced glycolysis in mouse model.
-The authors state in the materials and methods that the third group of rats is treated for 16 weeks with silica and then for the next 8 with Ac-SDKP. In contrast, macrophages in vitro, are pretreated with Ac-SDKP and then treated with SiO2? Why do the authors decide to pretreat macrophages in vitro?
In Figure 5, the authors compare the efficacy of Ac-SDKP with Ac-ADKP. The latter is to be described in the results or materials and methods.
In Figure 6C, the authors show the efficacy of three silencers for LDHA. This panel should be described in the results or materials and methods. Which silencer is chosen for the following experiments, number 2 or 3?
In Figure 7C, in the western blot legend the authors reported 32w, whereas in the manuscript and materials and methods they report a full treatment of 24w. Please, correct this inconsistency.
Author Response
Dear reviewer
Thank you very much for your comments concerning our manuscript, originally titled “Glycolytic reprogramming in silica-induced lung macrophages and silicosis reversed by Ac-SDKP treatment”. All comments were valuable and very helpful for revising and improving our paper, and they have provided important guiding significance for our research. We have carefully considered all of the comments and revised our manuscript accordingly. We hope that the changes we have made will meet with your approval. The revisions to the manuscript were marked up using the “Track Changes” function.
Point 1: the abstract is not clear, acronyms are not necessary, the cell experiments are in vitro and not in vivo, and the results could be enriched by a few more results, e.g. be less general.

Response 1: Thanks for your suggestion. We have revised the Abstract section. This error has been corrected in the revised manuscript.
Point 2: The authors introduce the mouse silicosis model, but on the same they do not evaluate the effect of Ac-SDKP on glycolysis enzymes, nor the switch to an M2 phenotype of macrophages. What is the value added of this model?
Response 2: In the present study, we used two different silicotic models to show the abnormal glycolysis reprogramming. The silicotic rats were made by inhale of silica. The silicotic mice were made by instillation of one-time dose of silica. All the silicotic model showed abnormal activation of key glycolytic enzymes.
Point 3: Line 235-236: Thus, the statement” We found that Ac-SDKP treatment attenuates glycolysis in silicotic rats and mice or in silica-induced NR8383 236 alveolar macrophages.” is incorrect, Ac-SDKP was not used to treat the silica-induced glycolysis in mouse model.
Response 3: Thank you for carefully reviewing my article. This error has been corrected in the revised manuscript.
Point 4: The authors state in the materials and methods that the third group of rats is treated for 16 weeks with silica and then for the next 8 with Ac-SDKP. In contrast, macrophages in vitro, are pretreated with Ac-SDKP and then treated with SiO2? Why do the authors decide to pretreat macrophages in vitro?
Response 4: The reviewer’s recommendation is very useful, and we will design the experiment for pre-treatment and post-treatment of Ac-SDKP in vitro.
Point 5: In Figure 5, the authors compare the efficacy of Ac-SDKP with Ac-ADKP. The latter is to be described in the results or materials and methods.
Response 5: We used Ac-ADKP, a functionally inactive analogue of Ac-SDKP (replacement of Ser by Ala in Ac-SDKP), as a control to evaluate the effects of Ac-SDKP. The related context in result and method sections has been revised.
Point 6: In Figure 6C, the authors show the efficacy of three silencers for LDHA. This panel should be described in the results or materials and methods. Which silencer is chosen for the following experiments, number 2 or 3?
Response 6: In the present study, we found that Ldha-siRNA #3 had the most significant inhibitory effect on LDHA protein level. Therefore, Ldha-siRNA #3 was used to silence LDHA in subsequent experiments. We have revised the detail in the Results section.
Point 7: In Figure 7C, in the western blot legend the authors reported 32w, whereas in the manuscript and materials and methods they report a full treatment of 24w. Please, correct this inconsistency.
Response 7: Thank you for carefully reviewing my article. This error has been corrected in the revised manuscript.
Round 2
Reviewer 1 Report
All comments have been addressed.